# Pro-Osteogenic Properties of *Violina pumpkin* (*Cucurbita moschata*) Leaf Extracts: Data from In Vitro Human Primary Cell Cultures

**DOI:** 10.3390/nu13082633

**Published:** 2021-07-30

**Authors:** Elisabetta Lambertini, Letizia Penolazzi, Giulia Pellielo, Caterina Pipino, Assunta Pandolfi, Serena Fiorito, Francesco Epifano, Salvatore Genovese, Roberta Piva

**Affiliations:** 1Department of Neuroscience and Rehabilitation, University of Ferrara, 44121 Ferrara, Italy; elisabetta.lambertini@unife.it (E.L.); maria.letizia.penolazzi@unife.it (L.P.); giulia.pellielo@edu.unife.it (G.P.); 2Department of Medical, Oral and Biotechnological Sciences, Center for Advanced Studies and Technology—CAST, University G. d’Annunzio of Chieti-Pescara, 66100 Chieti, Italy; c.pipino@unich.it (C.P.); pandolfi@unich.it (A.P.); 3Department of Pharmacy, University “G. D’Annunzio” of Chieti-Pescara, 66100 Chieti, Italy; serena.fiorito@unich.it (S.F.); francesco.epifano@unich.it (F.E.); salvatore.genovese@unich.it (S.G.)

**Keywords:** pumpkin leaf extracts, osteoblasts, osteoclasts

## Abstract

Traditional medicines rely mainly on use of plant extracts to mitigate or treat a wide range of disorders, including those that affect skeletal homeostasis. In this study, we investigated for the first time the potential pro-osteogenic effects of hexane, acetone and methanol extracts of the leaves of Cucurbita moschata, a very popular pumpkin cultivar in Western countries. We found that in Cucurbita moschata leaves, there are acetone-extractable substances—in particular, fatty acids such as 13-OH-9Z,11E,15E-octadecatrienoic acid (PU-13OH-FA), which is capable of both stimulating the function of human primary osteoblasts, which are responsible for bone formation, and inhibiting the differentiation of human osteoclasts, which are responsible for bone resorption. This dual effect was monitored by analyzing Runx2 expression, deposition of mineralized matrix, ALP activity, TRAP and actin ring staining respectively. This study suggests that bioactive chemicals from Cucurbita moschata leaves are potentially suitable as therapeutics for managing metabolic bone disorders such as osteoporosis and rheumatoid arthritis, and promoting tissue healing and functional recovery after bone fractures. The data we obtained increase knowledge on the biological activities of Cucurbita moschata, and in particular underline the potential benefits of consuming leaves which are a part of the plant currently little considered in the Western world.

## 1. Introduction

The nutritional benefits of pumpkin (*Cucurbita* sp.) have been widely demonstrated [1,2]. Pumpkin is a great source of potassium and beta-carotene, and contains large amounts of bioactive compounds that potentially contribute to the numerous health-promoting properties of the plant, including the strengthening of the immune system and counteracting the harmful effects of free radicals, thereby helping with the prevention of the development of metabolic syndrome, hypertension, atherosclerosis and cancer [1,3]. Various studies have demonstrated considerable diversity among the pumpkin cultivars in terms of their bioactive compounds, particularly concerning flavonoids, phenolic acids, tocopherols, minerals and vitamins [4,5]. All this evidence suggests the possibility of optimizing consumption for personalized diets or for specific health needs, by choosing the right pumpkin variety based on the nutrients supplied.

In most cases all organs of the pumpkin are edible and usable for obtaining herbal medicines; however, pulp, seeds and flowers are the most consumed parts all over the world, and those most investigated from a nutraceutical and therapeutic point of view [1,3].

Since it emerged that the consumption of the leaves as a food is practiced in some countries of the world, such as Nigeria, Ghana, Tanzania, Korea and India, with a positive impact on human health, scientists have also begun to take an interest in this part of the plant, which in the Western world, is instead considered essentially a waste product. Traditional medicine has in fact ascribed to pumpkin leaves some healing properties, such as hepatoprotective, antidiabetic and anticancer properties; antimicrobial, anti-inflammatory and antioxidant activity; and hematological improvement [6,7,8,9,10]. In order to better define their pharmacological potential, some research groups are working on phytochemical screening of the contents of leaf extracts from different cultivars. To date it has been shown that phosphorus; calcium; zinc; iron; copper; saponins; alkaloids; tannins; phenolics; vitamins A, K and C and folate; and various aminoacids and fatty acids are present [1,2,3,5,11].

The data available to date have not revealed evidence on the potential of pumpkin leaf extracts for preventing or slowing down diseases affecting the bone tissue.

A large body of evidence has been collected to show the potential of plant derivatives for the prevention and treatment of bone lesions [12,13,14]. Despite various discrepancies in the literature, there is agreement in attributing the positive effects of plant extracts, including those of the pumpkin tendrils, on the bone repair process, to antioxidant and anti-inflammatory action, and to the ability to increase angiogenesis and cell proliferation and differentiation [13,15,16,17]. Consistently with those observations, the few characteristics of pumpkin leaf extracts known prior to our study suggested that it would be reasonable to investigate their effects on bone cells. This also means scientifically supporting those benefits, including bone strengthening, for which regular consumption of pumpkin leaves is recommended according to popular traditions of some countries.

In the present study, we focused on a pumpkin cultivar that is very widespread in Western countries, the Violina pumpkin (Cucurbita moschata) [2], with the aim of discovering new extracts or functional food that can aid the slowing of bone loss and the functional restoration of damaged bone tissue. Hexane, acetone and methanol extracts of Violina pumpkin leaves were assessed on human primary osteoblasts. The beneficial effects of the acetone extract, and TLC subfraction 2 and the major compound isolated from it, an unsaturated hydroxy fatty acid, were discovered. The same acetone extract and its derivatives inhibited osteoclast differentiation, suggesting that extracts from Violina pumpkin leaves could be novel candidates for the prevention and treatment of bone disorders through their dual effects on osteoblast and osteoclast differentiation.

## 2. Material and Methods

### 2.1. Reagents and Chemicals

Ascorbic acid-2-phosphate, β-glycerophosphate, dexamethasone, 3-[4,5-dimethylthiazol-2-yl]-2,5-diphenyltetrazolium bromide (MTT), P-nitrophenyl phosphate (pNPP), Alizarin Red S (AR-S), paraformaldehyde, Triton X-100, fetal calf serum (FCS), L-glutamine and antibiotics (penicillin and streptomycin) were purchased from Sigma-Aldrich (Saint Louis, MO, USA). Alexa Fluor 488 Phalloidin (#A12379) was purchased from Thermo Fisher Scientific (Waltham, MA, USA). Antibody against human Runx2 (#sc-10758) was purchased from Santa Cruz Biotechnology (Dallas, TX, USA). Dulbecco’s modified eagle’s medium high glucose (DMEM), Ham’s F12 and phosphate-buffered saline (PBS 1X) were purchased from Euroclone (Milan, Italy). All organic solvents, HPLC-grade (methanol, acetone, n-hexane, dichloromethane), were supplied from Carlo Erba Reagents (Dasit Group Carlo Erba Reagenti, Milan, Italy). TLC plates (Analtech Uniplate, Silica gel GF, 20 × 20 cm, 500 microns) for preparative chromatography were purchased from Merck Sigma-Aldrich (Merck Sigma-Aldrich, Milan, Italy). Cromatographic column was performed on silica gel using column chromatography (Merck Sigma-Aldrich 60, 70-230 mesh, Fluka^®^ silica gel).

### 2.2. Sample Collection and Extraction Procedure

Fresh leaves of Cucurbita moschata were harvested in June 2020 at Alessandra Greco’ farm (Agriturismo “Alla Casella” Fondo Signa—Ca’ Coadi, 44124—Porotto FERRARA, Italy). The leaves were washed, air dried for a period of two weeks, cut into small pieces, homogenized to fine powder and then stored in vacuum bags.

Twenty grams of grounded raw material was extracted with 400 mL of each of the following solvents: *n*-hexane, acetone and methanol. Extractions were accomplished by three consecutive macerations (each extraction procedure at room temperature for 48 h) to obtain hexane (Hex), acetone (Ac) and methanol (Me) extracts as the final products. The first maceration process was carried out using *n*-hexane, the resultant extraction mixture was filtrated and the extracted solution was evaporated to dryness under vacuum. The residual plant material was resuspended in acetone, treated for 48 h at room temperature followed by filtration and evaporation of the filtrate. Finally, the same process was repeated in methanol. The extract yields (%) were 2.4%, 3.4% and 13.28% for hexane, acetone and methanol extracts, respectively. The organic extracts were weighted and dissolved in neat dimethyl sulfoxide (DMSO) to final concentrations of 50 mg/mL for hexane and methanol extracts and 30 mg/mL for acetone extracts. The concentrations of the stock solutions varied for different extracts, depending on their solubility. The stock solution was serially diluted with growth medium for the different assays. The final concentration of DMSO in cell culture medium in any experiment did not exceed 0.5%.

### 2.3. Thin Layer Chromatography (TLC) and Chromatographic Column

The preparative TLC on the pharmacologically performing acetone raw extract (Ac) was used firstly to separate and isolate enriched samples. The amount of the acetone extract processed was 600 mg. 

A preliminary screening was carried out on “normal TLC” to find the best condition in terms of mobile phase which was reported on the preparative TLC as a mixture of dichloromethane and methanol in a ratio of 97.5:2.5% for the final volume of 200 mL.

The Ac extract was resuspended in small amounts of the solvent mixture used for the mobile phase and applied as spots on the long streak. After a complete run, specific components were recovered by scraping the sorbent layer from the plate in the region of interest and eluting the separated material from the sorbent layer using a mixture of methanol and dichloromethane, followed by filtration and centrifugation to precipitate the silica gel. The supernatant was recovered and evaporated under vacuum. Three fractions of different polarity were recovered in acceptable amounts and identified as fractions 2, 6 and 7.

Fraction 2 underwent additional fractionation accomplished by silica-gel column chromatography. Dichloromethane was used as the sole solvent for a more accurate isolation of the pure fraction 2.

The pure fraction was characterized by ^1^H NMR at 25 °C on a 300 MHz Varian Oxford spectrometer, with CDCl_3_ as the solvent and chemical shifts in parts per million (δ) downfield from the internal standard TMS and by gas chromatography-mass spectrometry (GC-MS) analysis using a GC-MS apparatus (8860 GC with 5977B GC/MSD Agilent system) after derivatization with a methyl ester obtained by treatment with CH_2_N_2_. The value of purity for the compound present in fraction 2 was 99.7%. Analytical data are in full agreement with those already reported in the literature for the same compound determined to be 13-OH-9Z,11E,15E-octadecatrienoic acid [18]. Fractions 6 and 7 were further purified by crystallization from H_2_O, providing ferulic and p-cumaric acid, respectively (see Appendix A Section for analytical data).

### 2.4. Cell Isolation and Culture 

Human osteoblasts (hOBs) were obtained from vertebral lamina discarded during spinal surgery. Bone fragments were obtained from 10 donors (mean age 63 years) using a research protocol approved by the Ethics Committee of the University of Ferrara and S. Anna Hospital (protocol approved on 17 November 2016). Bone chips were minced into smaller pieces as previously reported [19], plated in T-25 culture flasks (Sarstedt, Nümbrecht, Germany) and cultured in basal medium (50% DMEM high-glucose/50% Ham’s F12, 10% FCS). The cells were expanded until confluent (passage zero, P0), harvested and used for further experiments (passage 1 to passage 3). 

For peripheral blood mononuclear cells (PBMCs), healthy volunteers (*n* = 3, median age 47 years) were recruited after informed consent, and then PBMCs were obtained from diluted peripheral blood (1:2 in Hanks solution) which was separated by Histopaque^®^-1077 (Sigma- Aldrich) as previously described [20]. Briefly, Monocytes (hMCs) were purified from PBMCs by adhesion selection on polystyrene plates: 1 × 10^6^ PBMCs/cm^2^ were plated, allowed to settle for 4 h at 37° and flasks were then rinsed to remove non-adherent cells. Osteoclast differentiation from isolated hMCs was induced by adding 25 ng/mL of M-CSF and 30 ng/mL of RANKL (PeproTech EC Ltd., London, UK) in culture medium, and after 14 days, TRAP staining was carried out.

### 2.5. Cell Viability 

To evaluate cell viability, hOCs and hOBs were seeded into 96-well plate and treated with increasing concentrations of the extracts/fractions/purified compound (5–250 μg/mL) or DMSO alone (0.5%), and incubated for 72 h. At the end of the treatment, cells were washed with PBS 1X and incubated with 0.5 mg/mL of 3-(4,5-cimethylthiazol-2-yl)-2,5-diphenyl tetrazolium bromide (MTT) reagent in culture medium for an another 3 h. After incubation, 100 µL DMSO was added to each well to dissolve the formazan crystals, and the optical density at 570 nm was measured with a microplate reader (Sunrise™ Absorbance Reader, Tecan Group Ltd., Männedorf, Switzerland). Cell viability was determined as a percentage compared to the control. The experiments were repeated three times, and triplicates were done for each condition in each assay.

For Calcein AM/propidium iodide (PI), cells were seeded in 24-well plates at a density of 10.000 cells/well, and treated with hexane, acetone or methanol extracts (5 μg/mL) for 72 h. Before staining, the medium was removed from the wells, and 500 μL of the staining solution was added to each well. The samples were incubated in the dark at room temperature for 15 min, and thereafter the wells were rinsed with PBS 1X and immediately visualized under a fluorescence microscope (Nikon Eclipse 50i, Nikon Corporation, Tokyo, Japan). Dead cells stained red; viable ones appeared green.

### 2.6. F-Actin Ring Immunofluorescence Assay

For analysis of cellular morphology, actin ring formation and organization, hOBs and hOCs were fixed with 4% paraformaldehyde (PFA) for 2 min and permeabilized with 0.2% Triton X-100 in PBS 1X for 15 min. The cells were then stained with Alexa Fluor 488 Phalloidin (1:500 dilution in PBS 1X) at room temperature in the dark for 30 min. The cells were washed with PBS 1X, and the nuclei were counterstained with DAPI solution (Sigma Aldrich). Fluorescent images were obtained with a fluorescence microscopy (Nikon Eclipse 50i).

### 2.7. In Vitro Scratch-Wound Healing Assay

The ability of hOBs to migrate to the wounded area was assessed using the scratch assay method [21]. Cells were seeded into 24-well plates and incubated with complete medium at 37 °C and 5% CO_2_. After 24 h of incubation, the monolayer confluent cells were scrapped horizontally with a sterile 200 µL pipette tip. The debris was removed by washing with PBS 1X. Then the cells were treated with 5 µg/mL of hexane, acetone or methanol extracts of Pumpkin leaves. The cells treated with DMSO were used as a control. The scratch that represented a wound was photographed using an inverted microscope (Nikon Corporation, Tokyo, Japan) equipped with a digital camera, at 0 h. After 24–48 h of incubation with extracts, the other set of images was photographed. To determine the migration rate, the images were analyzed using “ImageJ” software, and the percentage of the closed area was measured and compared with the value obtained at 0 h. An increase in the percentage of the closed area indicated the migration of cells. Experiments were performed in triplicate. Wound closure (%) = (measurement at 0 h − measurement at 24/48 h)/measurement at 0 h × 100. 

### 2.8. Immunocytochemistry

Immunocytochemistry analysis was performed by employing the ImmPRESS (#MP-7500; Vectorlabs, Burlingame, CA, USA). Cells grown in culture plates were fixed in cold 100% methanol and permeabilized with 0.2% (*v*/*v*) Triton X-100 (Sigma Aldrich, St. Louis, MO, USA) in TBS 1X (Tris-buffered saline). Cells were treated with 3% H_2_O_2_ in TBS 1X, and incubated in 2% normal horse serum (Vectorlabs) for 15 min at room temperature. After the incubation in blocking serum, polyclonal antibody against Runx2 (M-70 #sc-10758—rabbit antihuman, 1:200 dilution, Santa Cruz Biotechnology) was added, and the plate was incubated overnight (4 °C). After rinsing in TBS 1X, the cells were incubated for 30 min at room temperature with ImmPRESS reagent and then stained with substrate/chromogen mix (ImmPACT™ DAB). After washing, the cells were mounted in glycerol/PBS (9:1) and observed with a Nikon Eclipse 50i optical microscope. Quantitative image analysis of immunostained cells was performed using ImageJ software as previously reported [22]. 

### 2.9. Alkaline Phosphatase (ALP) Activity Assay

hOBs were plated at a density of 20,000 cells/cm^2^ in 24-well plates and exposed to extracts or fractions or purified compound in osteogenic medium (OM) (DMEM high-glucose, 10% FCS supplemented with 10 mM β-glycerophosphate, 100 nM dexamethasone and 100 μM ascorbate) for 7 or 14 days. Control group cells were cultured in OM containing 0.5% DMSO. Extracts containing media were replaced twice weekly. Alkaline phosphatase (ALP) activity was measured by analyzing the rate of p-nitrophenyl phosphate disodium hexahydrate (pNPP) hydrolysis.

At each time point, cells were washed with PBS 1X twice and lysed with 0.1% Triton X-100 for 20 min on ice. After centrifugation at 14,000 rpm for 5 min at 4 °C, supernatants were collected and stored at –80°. Upon defrosting, 50 µL of lysate was incubated with 50 µL of pNPP solution containing 2 mg/mL pNPP substrate in buffer (1 M Diethanolamine; 0.5 mM MgCl_2_, pH 10.3) for 30 min at 37 °C. The enzymatic reaction was stopped through the addition of 100 μL of 0.1 M NaOH, and the absorbance was measured at 405 nm by using a microplate reader (Sunrise™ Absorbance Reader). Finally, ALP activity was normalized to protein content, determined by Bradford assay (Sigma) at OD595. 

### 2.10. Alizarin Red S Staining (AR-S)

hOBs were seeded at density of 20,000 cells/cm^2^ in 12-well plates. At confluence, cells were exposed to extracts or fractions or purified compound in OM for 21 days. Control group cells were cultured in OM containing 0.5% DMSO. During osteogenic differentiation, culture medium was replaced twice weekly. At the end of treatment, cells were subjected to Alizarin Red S (AR-S) staining. Cells were fixed with 70% *v/v* cold ethanol for 1 h and stained with 40 mM AR-S (pH 4.1) at room temperature for 20 min with gentle agitation. The staining solution was discarded, and cells were washed five times with H_2_O and two times with cold PBS 1X before being left to air dry. Mineralized ARS-positive nodules present in each well were visualized using an inverted microscope. For the quantification of mineralization, AR-S was extracted with 10% cetylpyridinium chloride (CPC) in sodium phosphate for 1 h, followed by absorbance measurement at 570 nm by using a microplate reader (Sunrise™ Absorbance Reader).

### 2.11. Tartrate-Resistant Acid Phosphatase (TRAP) Staining 

TRAP staining of the cells was performed as previously reported [20]. Briefly, the cells were fixed in 4% paraformaldehyde (PFA) with 0.1 M cacodilic buffer, pH 7.2 (0.1 M sodium cacodilate, 0.0025% CaCl_2_) for 15 min, extensively washed in the same buffer, and stained for TRAP (Acid Phosphatase Leukocyte Kit) according to the manufacturer’s protocol. After washing with distilled water and drying, mature TRAP positive multinucleated cells containing more than three nuclei were counted as osteoclasts. 

### 2.12. Statistical Analysis

Statistical analysis was conducted using GraphPad Prism 8.0 (GraphPad Software, San Diego, CA, USA). Mean and standard deviation values (mean ± SD) were calculated for all statistically analyzed parameters. The differences between groups were analyzed using analysis of variance (ANOVA) followed by Tukey’s post-hoc test. *p* < 0.05 was considered statistically significant.

## 3. Results 

### 3.1. Effects of Hexane, Acetone and Methanol Leaf Extracts on Osteoblast Viability, Morphology and Migration

To test the cytotoxicity, hOBs were treated with hexane, acetone and methanol extracts of the leaves of Cucurbita moschata (5, 10, 50, 100 and 250 μg/mL) for 72 h, and cell viability was investigated by using MTT assay. As shown in Figure 1A, only the highest concentration of acetone extract was found to significantly affect cell viability. For the next experiments, a concentration of 5 μg/mL was used. Staining with Calcein AM/propidium iodide (PI) (Figure 1B) and FITC-conjugated phalloidin (Figure 1C) confirmed that cells were viable and maintained normal cytoskeletal organization when treated with hexane, acetone and methanol extracts at concentrations of 5 μg/mL. 

The in vitro scratch assay was performed to detect the influence of each leaf extract on the cell migration at 24 and at 48 h. As shown in Figure 2, the treatments with hexane, acetone and methanol extracts did not affect the ability of hOBs to migrate. In fact, there was no significant difference in the rate of gap closure of the cells treated with the leaf extracts compared to control cells.

### 3.2. Effects of Hexane, Acetone and Methanol Leaf Extracts on Osteogenic Markers

The basal expression level of the master regulator of osteogenic differentiation, Runx2 transcription factor, was analyzed after 72 h of hOB exposure to each leaf extract without osteogenic inducers. Interestingly, as shown in Figure 3A, the treatment with acetone extract significantly increased Runx2 expression compared with vehicle-treated cells. On the contrary, treatments with hexane and methanol extracts did not significantly affect Runx2 expression.

In order to investigate the effect of each leaf extract on osteoblast differentiation, functional assays on hOBs cultured in osteogenic medium were then performed by evaluating ALP activity (Figure 3B) and mineral matrix deposition (Figure 3C,D). As shown in Figure 3B, after 7 days in culture, the presence of an extract did not significantly affect ALP activity. After 14 days in culture, acetone extract-treated cells showed significantly higher ALP activity than the hexane and methanol extract-treated cells and vehicle-treated cells (Figure 3B). For what concerns the mineral matrix deposition, acetone extract-treated cells showed significantly higher Alizarin Red S staining than the hexane and methanol extract treated cells and vehicle-treated cells (Figure 3C,D).

### 3.3. The Fractionation of the Acetone Extracts and the Osteogenic Potential of TLC Subfractions

In light of the above results, the acetone extract was analyzed further in order to identify the potential active compounds contained in this mixture. Therefore, the acetone extract was purified by preparative thin-layer chromatography (TLC) and three of the seven TLC subfractions (the most abundant, 2, 6 and 7; see Appendix A) were then recovered and tested. For what concerns the cell viability, hOBs were treated with the three subfractions (concentrations 5, 10, 50, 100 and 250 μg/mL) for 72 h, and then subjected to MTT assays. As shown in Figure 4A, only the highest concentrations of the subfractions affected cell viability. For the next experiments, concentration 5 μg/mL was used.

With the same protocol adopted for the raw acetone extract, the osteogenic potential of the TLC subfractions was evaluated. All three subfractions significantly increased Runx2 basal expression levels with respect to the raw acetone extract and untreated cells (Figure 4B). The highest effect was observed for subfraction 2. When functional assays were performed, the highest ALP activity and mineral matrix deposition were exhibited by subfraction 2-treated hOBs (Figure 4C,D). 

Subfraction 2 was then subjected to additional fractionation accomplished by silica-gel column chromatography. The major compound was isolated, purified and identified by ^1^H NMR and by gas chromatography-mass spectrometry (GC-MS) as the unsaturated hydroxy fatty acid 13-hydroxy-9Z,11E,15E-octadecatrienoic acid (PU-13OH-FA), which was already previously identified in the leaves of Cucurbita moschata [18]. The chemical structure and mass spectrum of the compound are reported in Appendix A. The effects of PU-13OH-FA were then tested on the hOBs (Figure 5). We found that the PU-13OH-FA alone significantly affected cell viability far more than the subfraction 2. As shown in Figure 5A, the decrease in cell viability was significantly greater in PU-13OH-FA-treated hOBs than in subfraction 2-treated hOBs in the concentration range 50–250 μg/mL. Additionally, for this single compound the concentration 5 μg/mL was used for investigating its biological activity. Interestingly, the treatment with PU-13OH-FA significantly increased Runx2 basal expression, ALP activity and mineral matrix deposition compared to the respective control, but said effects never surpassed those of the subfraction 2 treatment (Figure 5B–E). Characterizations of the compounds present in fractions 6 and 7 are reported in Appendix A.

### 3.4. Acetone Extracts and Osteoclastogenesis 

We then moved to investigate the effects of acetone extract, TLC subfraction 2 (Fr2) and the PU-13OH-FA compound on osteoclastogenesis. Human primary monocytes from peripheral blood were cultured in the presence of osteoclastogenic inducers together with acetone extract, or Fr2 or the single PU-13OH-FA compound. As shown in Figure 6A, only the highest concentrations of acetone extract (100–250 μg/mL) and the treatment with 250 μg/mL of PU-13OH-FA significantly decreased cell viability. To evaluate the effects of the acetone extract, Fr2 and PU-13OH-FA on osteoclastogenesis, TRAP staining was performed. We evaluated the cells with ≥3 nuclei. The results showed that Fr2 did not affect tartrate-resistant acid phosphatase (TRAP) activity. The number of TRAP-positive multinucleated cells was significantly decreased in the presence of acetone raw extract or PU-13OH-FA (Figure 6B). We next determined the effects of the treatments on the actin ring formation, which is crucial for bone resorption by osteoclasts [23]. As shown in Figure 6C, acetone extract, Fr2 and the single PU-13OH-FA compound significantly decreased actin ring structures in comparison with control osteoclasts. In this case, the most effective was the acetone extract.

## 4. Discussion 

The interest in natural products for bone healing and functional restoration of damaged bone tissue following trauma, fractures, metabolic and congenital diseases is increasing [12,24]. Studies demonstrating the positive effects of plant extracts/compounds on bone matrix formation and cellular activity are opening up a new perspective on the development of new promising therapy as an alternative to conventional medicine and synthetic products, and drawing attention to appropriate food consumption [25,26]. However, it is important to consider that to date, the mechanisms by which compounds of natural origin act in the bone remodeling supported by osteoblasts and osteoclasts and consequently in bone healing are still poorly understood [12,13,15,24]. In this regard, the literature contains numerous conflicting data that point out the need to improve experimental designs and methodological procedures, together with the adequacy of the experimental models as means to ensure an adequate level of scientific evidence.

In the present study we used human primary cultures of osteoblasts and osteoclasts to demonstrate for the first time, to the best of our knowledge, the potential bone health benefits of extracts from a little investigated part of pumpkin plants, the leaves. In most Western countries, pumpkin leaves are considered waste products and are not used in food. In some countries, such as Nigeria, Ghana, Tanzania, Korea and India, the consumption of pumpkin leaves is practiced, sometimes for therapeutic use according to the indications of traditional medicine [1,2,3,4,5,6,7,8,9,10].

Here, we used the leaves of Violina pumpkin (Cucurbita moschata, a pumpkin cultivar that is very widespread in Western countries) and hexane, acetone and methanol as solvents to extract potential bioactive compounds.

The biological activity of the extracts was tested on human primary osteoblasts in terms of expression of osteogenic markers, formation of mineralization nodules and cell migration ability [27]. The extract obtained from acetone exhibited a high capacity for inducing osteogenic differentiation, suggesting that the cells can benefit from the mixture present in this extract. These pieces of evidence led us to purify the acetone extract and evaluate the osteogenic potential of its TLC subfractions, among which we found one (named subfraction 2) that, more than the others, showed significant activity. The characterization of this fraction showed that it contains almost exclusively the unsaturated hydroxy fatty acid 13-hydroxy-9Z,11E,15E-octadecatrienoic acid (PU-13OH-FA). This molecule was already isolated from the leaves of Cucurbita moschata [18], but its biological properties have never been studied. Here we found that PU-13OH-FA treated-osteoblasts significantly increased Runx2 basal expression level, ALP activity and mineral matrix deposition compared to untreated cells, but always less than subfraction 2—treated cells. These pieces of evidence suggest that the components of the mixture present in subfraction 2, even if present in very low doses, can exert a synergistic effect in relation to the investigated pro-osteogenic properties. This is in agreement with numerous pieces of in vitro and in vivo evidence demonstrating that the biological activity of plant-derived products is generally due to the synergism between its constituents [13]. Therefore, when the effectiveness of the whole plant extract is higher than that of the isolated compounds, it means that the synergistic behavior of multiple constituents prevails over the antagonistic interaction which, conversely, could favor the potential therapeutic application of a single purified compound [13,28].

Interestingly, when the same acetone extract and its derivatives were tested on human primary osteoclasts, the differentiation significantly decreased. Therefore, as a whole our data suggest that in Cucurbita moschata leaves there are acetone-extractable substances, in particular, fatty acids such as PU-13OH-FA, capable of both stimulating the function of osteoblasts, which are responsible for bone formation, and inhibiting the differentiation of osteoclasts, which are responsible for bone resorption [29]. This dual effect suggests that the bioactive chemicals from Cucurbita moschata leaves are potentially suitable as ideal therapeutics both in managing metabolic bone disorders such as osteoporosis and rheumatoid arthritis, and in promoting tissue healing and functional recovery after a bone fractures. Although more studies are needed, our data suggest that consuming leaves may be nutritionally beneficial for bone health. Even if the scenario linking fatty acids to bone health is complex and the debate remains controversial, undoubtedly important roles of fatty acids in the metabolism of bone tissue have long been studied and different potential action mechanisms have been investigated [30,31,32,33]. For example, there is large body of evidence that enriching diets with specific long-chain polyunsaturated fatty acids are associated with higher bone mass. Peak bone mass was increased in adolescents and bone loss was reduced in an animal model of osteoporosis [34]. It will be, therefore, useful to expand our knowledge on the nutritional properties of parts of plants, such as pumpkin leaves, which many countries of the world, incorrectly, do not consider edible. As a whole, our data can therefore be the premise for encouraging the consumption of pumpkin leaves in Western countries. However, the bone health-promoting properties of Cucurbita moschata leaves require, of course, further investigation, for different reasons that are given below. 

From nutraceutical and therapeutic points of view, it will be important to investigate the properties of the leaves from different pumpkin cultivars and also the correlations among leaves’ nutritional value and the characteristics of the soil or climate where the plants are grown.

Regarding how to use pumpkin leaves as a food, another important aspect will be to understand the changes in the chemical composition resulting from different cooking techniques to select the most suitable ones that can reduce the possible loss of bone health properties of pumpkin leaf extracts [35].

From the evidence we have so far gathered, it will be necessary to move to perform further molecular and cellular in vitro experiments in order to investigate the mechanisms supporting the effects on bone cells, and pre-clinical studies in vivo, in order to validate the in vitro data and to ensure that there are not unwanted side effects associated with the use of the pumpkin leaf extracts. It is important to underline that performing the experiments with human primary bone cell cultures certainly has added value over the use of cell lines, as they resemble the cell behavior in vivo. However, osteoblasts from bone biopsies and osteoclast precursors from blood samples are not common, and this reduces the number of experiments that can be carried out and requires caution. For this reason, further experiments will be carried out in the near future, including a more in-depth comparison of extraction techniques by establishing a more appropriate delivery vehicle, and an investigation into the potential osteo-protective effects of the extracts analyzed here in terms of anti-inflammatory and antioxidant properties [3,7,36,37]. 

Finally, considering that some skeletal lesions require the use of scaffolds, the data presented here could also be useful in the regenerative medicine field for the realization of the so-called “herbal scaffolds,” bone substitutes that are fabricated by combining bioactive plant extracts with suitable polymeric matrices to optimize cell migration, proliferation, differentiation and the balance between osteoblasts and osteoclasts [15].

## 5. Conclusions

In this study we described the effect of extracts from leaves of Cucurbita moschata in terms of stimulation of human osteoblast activity and inhibition of osteoclast differentiation. This dual effect has been mainly attributed to acetone-extractable substances; in particular, fatty acids such as 13-OH-9Z,11E,15E-octadecatrienoic acid (PU-13OH-FA). As a whole we suggest that bioactive chemicals from Cucurbita moschata leaves could be potentially useful in bone health, underlining the potential benefits of their dietary consumption.

## Figures and Tables

**Figure 1 nutrients-13-02633-f001:**
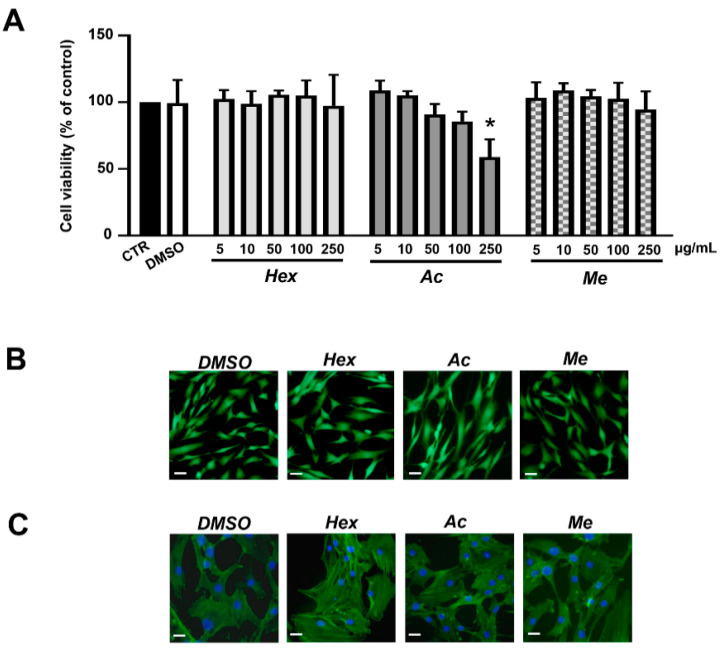
(**A**) Effects of organic extracts of Violina pumpkin leaves on cell viability. hOBs were treated with 5 –250 µg/mL of hexane (*Hex*), acetone (*Ac*) and methanol (*Me*) extracts for 72 h. The viability was monitored with MTT assays. Data represent mean ± SD (*n* = 3). * *p* < 0.05 vs. control cells (CTR). (**B**) Cell viability in the presence of extracts (5 µg/mL) was also evaluated by double staining with Calcein AM/propidium iodide. The green fluorescence indicates the presence of Calcein labeled live cells; propidium iodide-labeled dead cells are revealed by red fluorescence. Merged photomicrographs are reported. (**C**) Effects of organic extracts of Violina pumpkin leaves on F-actin cytoskeleton organization. hOBs were treated with Hex, Ac and Me extracts (5 µg/mL) for 72 h and cytoskeletal organization was analyzed by Alexa Fluor 488 Phalloidin staining and a fluorescence microscope (Nikon Eclipse 50i). Representative images of the cells are reported. Nuclei were counterstained with DAPI (blue). CTR = control untreated cells; DMSO = vehicle (DMSO 0.5%) treated cells. Scale bars: 50 µm.

**Figure 2 nutrients-13-02633-f002:**
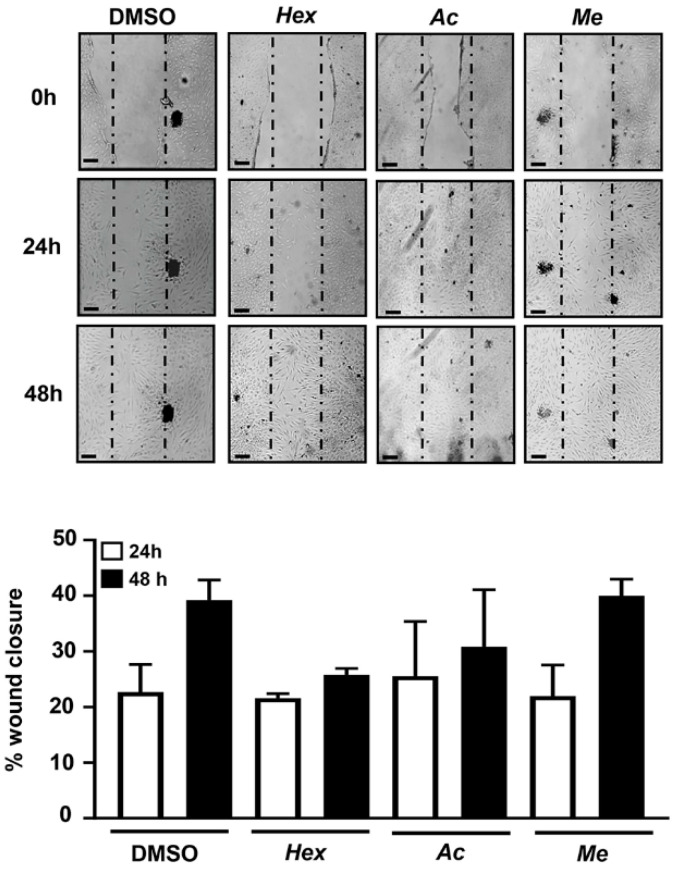
Effects of organic extracts of Violina pumpkin leaves on cell migration. The migration of hOBs was measured with wound scratch assays. hOBs were incubated in presence of Hex, Ac and Me extracts (5 µg/mL), and images were captured at 0, 24 and 48 h. The boundaries of the scratched wounds were determined by the dashed dark lines. The quantitative evaluation and statistical analysis of wound closure percentage were performed by ImageJ software. Results are expressed as means ± SD (*n* = 3). DMSO = vehicle (DMSO 0.5%) treated cells. Representative photomicrographs were taken after 0, 24 and 48 h into extract treatments. Scale bars: 50 µm.

**Figure 3 nutrients-13-02633-f003:**
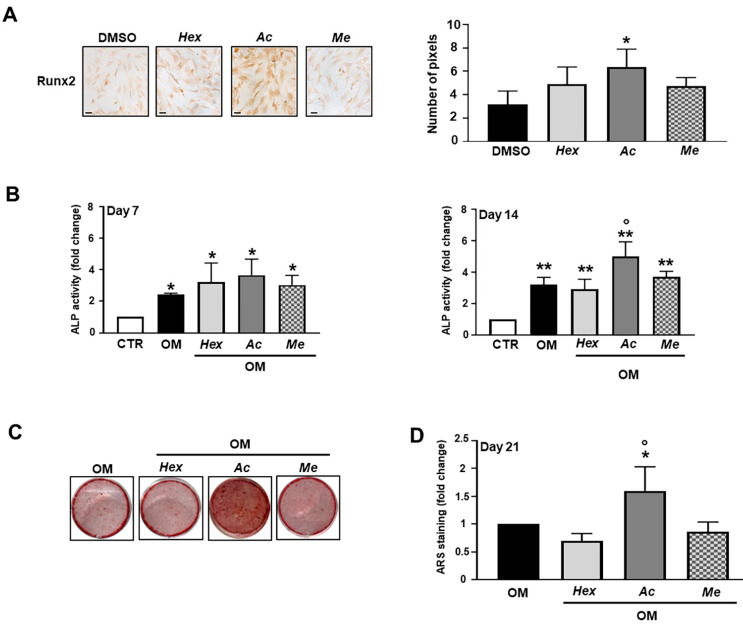
Effects of organic extracts of Violina pumpkin leaves on osteogenic markers. (**A**) The expression of Runx2 early osteogenic differentiation marker was evaluated. hOBs were maintained in basal medium and treated with 5 µg/mL of hexane (*Hex*), acetone (*Ac*) and methanol (*Me*) extracts for 72 h. Representative optical photomicrographs of immunostaining are reported. Scale bars: 20 µm. Protein levels were quantified by densitometric analysis of immunocytochemical pictures using ImageJ software and are expressed as means of pixels per one hundred cells ± SD (*n* = 3). * *p* < 0.01 vs. DMSO. (**B**) ALP activity was determined after 7 and 14 days of extract treatment in osteogenic medium. Data are presented as fold changes with respect to control group (CTR) (mean ± SD) (*n* = 3). * *p* < 0.05 vs. CTR, ** *p* < 0.01 vs. CTR, ° *p* < 0.05 vs. OM and Hex-treated groups. (**C**) Representative images of mineralized nodule formation detected using Alizarin Red S staining. Cells were cultured in osteogenic medium in presence of extracts for 21 days. (**D**) Quantitative analysis of Alizarin Red S staining was performed after extraction with cethylpyridinium chloride. Data are presented as fold changes with respect to OM (mean ± SD) (*n* = 3). * *p* < 0.05 vs. OM, ° *p* < 0.01 vs. Hex and Me-treated groups. CTR = cells maintained in basal medium containing 0.5% DMSO. OM = osteogenic medium containing 0.5% DMSO.

**Figure 4 nutrients-13-02633-f004:**
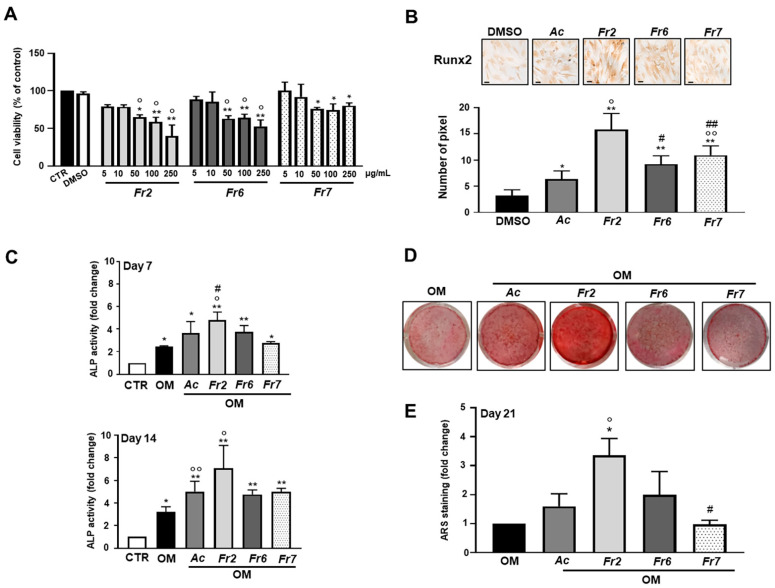
Effects of TLC subfractions of acetone raw extract on cell viability and osteogenic markers. (**A**) hOBs were treated with 5–250 µg/mL of TLC subfractions 2, 6 and 7 (Fr2, Fr6, Fr7) for 72 h. The viability was monitored with MTT assays. Data represent mean ± SD (*n* = 3). * *p* < 0.05 vs. control cells (CTR), ** *p* < 0.001 vs. CTR, ° *p* < 0.001 vs. DMSO-treated group. (**B**) The expression of Runx2 early osteogenic differentiation marker was evaluated. hOBs were maintained in basal medium and treated with 5 µg/mL of TLC subfractions or raw acetone extract for 72 h. Representative optical photomicrographs of immunostaining are reported. Scale bars: 20 µm. Protein levels were quantified by densitometric analysis of immunocytochemical pictures using ImageJ software and are expressed as means of pixels per one hundred cells ± SD (*n* = 3). * *p* < 0.05 vs. DMSO, ** *p* < 0.001 vs. DMSO, ° *p* < 0.001 vs. Ac, °° *p* < 0.01 vs. Ac, ^#^ *p* < 0.001 vs. Fr2, ^##^ *p* < 0.01 vs. Fr2. (**C**) ALP activity was determined after 7 and 14 days for TLC subfraction and raw acetone extract treatments in an osteogenic medium. Data are presented as fold changes with respect to the control group (CTR) (mean ± SD) (*n* = 3). * *p* < 0.05 vs. CTR, ** *p* < 0.01 vs. CTR, ° *p* < 0.01 vs. OM, °° *p* < 0.05 vs. OM, ^#^ *p* < 0.05 vs. Fr7-treated group. (**D**) Representative images of the formation of mineralized nodules detected using Alizarin Red S staining. Cells were cultured in osteogenic medium in the presence of TLC subfractions or raw acetone extract for 21 days. (**E**) Quantitative analysis of Alizarin Red S staining was performed after extraction with cethylpyridinium chloride. Data are presented as fold changes with respect to OM (mean ± SD) (*n* = 3). * *p* < 0.01 vs. OM, ° *p*< 0.01 vs. Ac-treated group, # *p* < 0.01 vs. Fr2-treated group. CTR = cells maintained in basal medium containing 0.5% DMSO. OM = osteogenic medium containing 0.5% DMSO.

**Figure 5 nutrients-13-02633-f005:**
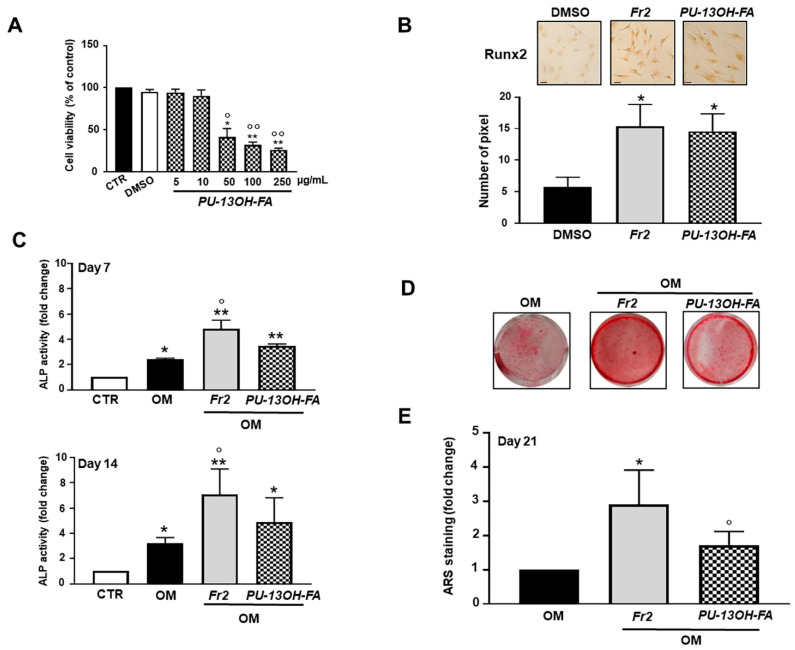
Effects of a pure compound (PU-13OH-FA) obtained from subfraction 2 on cell viability and osteogenic markers. (**A**) hOBs were treated with 5–250 µg/mL of compound PU-13OH-FA for 72 h. The viability was monitored with an MTT assay. Data represent mean ± SD (*n* = 3). * *p* < 0.05 vs. control cells (CTR), ** *p* < 0.001 vs. CTR, ° *p* < 0.05 vs. DMSO-treated group, °° *p* < 0.01 vs. DMSO-treated group. (**B**) The expression of Runx2 early osteogenic differentiation marker was evaluated. hOBs were maintained in basal medium and treated with 5 µg/mL of Fr2 and PU-13OH-FA for 72 h. Representative optical photomicrographs of immunostaining is reported. Scale bars: 20 µm. Protein levels were quantified by densitometric analysis of immunocytochemical pictures using ImageJ software and expressed as means of pixels per one hundred cells ± SD (*n* = 3). * *p* < 0.001 vs. DMSO. (**C**) ALP activity was determined after 7 and 14 days of Fr2 and PU-13OH-FA treatment in osteogenic medium. Data are presented as fold changes with respect to the control group (CTR) (mean ± SD) (*n* = 3). * *p* < 0.05 vs. CTR, ** *p* < 0.01 vs. CTR, ° *p* < 0.01 vs. OM. (**D**) Representative images of mineralized nodules formation detected using Alizarin Red S staining. Cells were cultured in osteogenic medium in the presence of Fr2 and PU-13OH-FA for 21 days. (**E**) Quantitative analysis of Alizarin Red S staining was performed after extraction with cethylpyridinium chloride. Data are presented as fold changes with respect to OM (mean ± SD) (*n* = 3). * *p* < 0.001 vs. OM, ° *p* < 0.05 vs. Fr2-treated group. CTR = cells maintained in basal medium containing 0.5% DMSO. OM = osteogenic medium containing 0.5% DMSO.

**Figure 6 nutrients-13-02633-f006:**
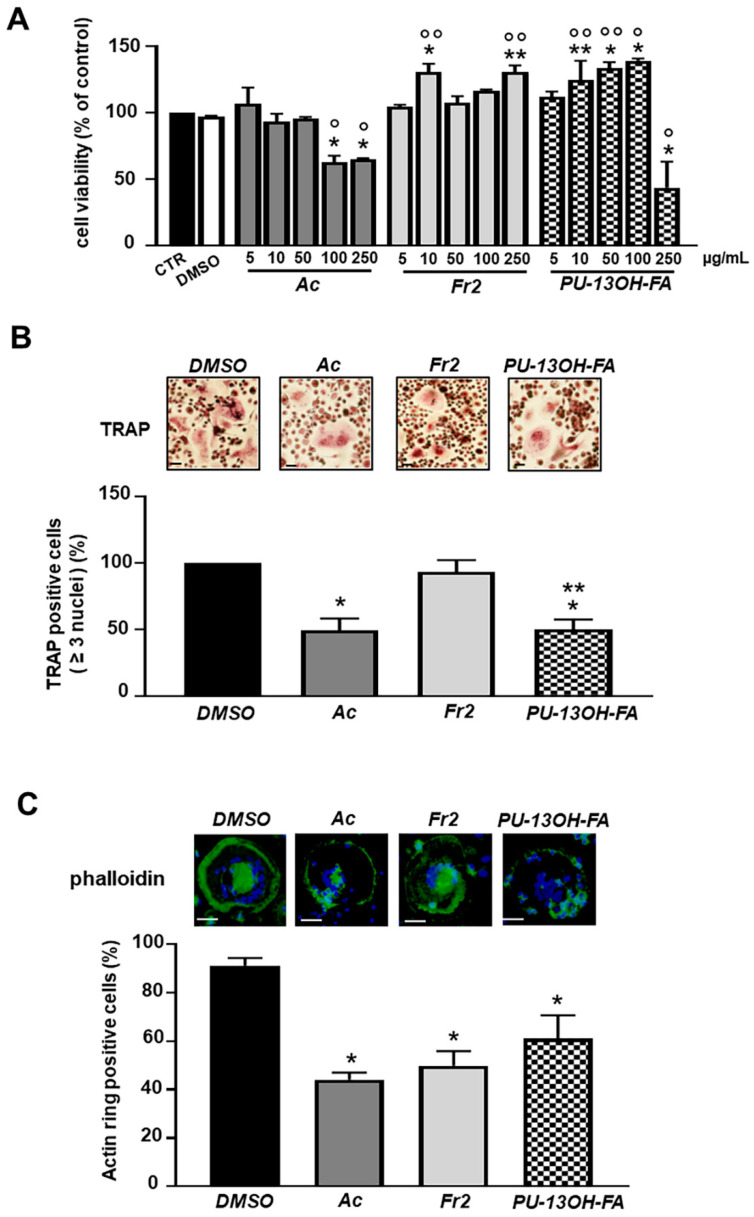
Effects of acetone extract (Ac), TLC subfraction 2 (Fr2) and pure compound (PU-13OH-FA) on cell viability and osteoclast differentiation. (**A**) Human osteoclasts (hOCs) were treated with 5–250 µg/mL of Ac, Fr2 or PU-13OH-FA for 72 h. The viability was monitored with MTT assay. Data represents mean ± SD (*n* = 3). * *p* < 0.001 vs. control cells (CTR), ** *p* < 0.05 vs. CTR, ° *p* < 0.001 vs. DMSO, °° *p* < 0.05 vs. DMSO-treated group. To assess the effects of acetone extract (Ac), TLC subfraction 2 (Fr2) and pure compound (PU-13OH-FA) on osteoclastogenesis, monocytes were cultured in osteoclastogenic medium in the presence of Ac, Fr2, PU-13OH-FA (5 µg/mL) or DMSO alone (0.5%) for 14 days. (**B**) After culturing for 14 days, cells were fixed and stained for TRAP, and representative images are reported. Scale bars: 50 µm. Graphic illustration of percentage of TRAP-positive cells (≥3 nuclei). Data represent mean ± SD (*n* = 3). * *p* < 0.001 vs. DMSO-treated group, ** *p* < 0.001 vs. Fr2. (**C**) hOCs actin rings were analyzed by phalloidin staining; nuclei were counterstained with DAPI. Scale bars: 50 µm. Data are presented as the percentages of actin ring-positive cells relative to the total number of osteoclasts, which were evaluated by two independent investigators in 10 randomly selected optical fields. Data are presented as the mean ± SD (*n* = 3). * *p* < 0.001 vs. DMSO-treated group.

## Data Availability

All datasets generated for this study are included in the article.

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
