# Peer review of "Pro-Osteogenic Properties of Violina pumpkin (Cucurbita moschata) Leaf Extracts: Data from In Vitro Human Primary Cell Cultures"

_nutrients, 2021, doi:10.3390/nu13082633_

Round 1

Reviewer 1 Report

1. The author identified the active substance of the leaf of Cucurbita mos-chata extract by looking at the function related to the promotion of bone formation.

2. Readers want to know what substances are in fractions 6 and 7. To refer to the activity of fractions 6 and 7, please identify the compounds in fractions 6 and 7.

3. Add the NMR and MS data of compound 2 to the Supplementary Materials.

4. Recently, research on Cucurbita mos-chata that is good for bones is continuously in progress (some mention in the introduction)

4-1. New 8-C-p-Hydroxylbenzylflavonol Glycosides from Pumpkin (Cucurbita moschata Duch.) Tendril and Their Osteoclast Differentiation Inhibitory Activities
By: Kim, Kiok; Choi, Joo-Hee; Oh, Jisu; Park, Ji-Yeon; Kim, Young-Min; Moon, Jae-Hak; Park, Jong-Hwan; Cho, Jeong-Yong
Molecules (2020), 25(9), 2077 | Language: English, Database: CAplus and MEDLINE

4-2. Dehydrodiconiferyl alcohol promotes BMP-2-induced osteoblastogenesis through its agonistic effects on estrogen receptor
By: Lee, Wonwoo; Ko, Kyeong Ryang; Kim, Hyun-keun; Lim, Seonung; Kim, Sunyoung
Biochemical and Biophysical Research Communications (2018), 495(3), 2242-2248 | Language: English, Database: CAplus and MEDLINE

4-3. Water extract of tendril of Cucurbita Moschata Duch. suppresses RANKL-induced osteoclastogenesis by down-regulating p38 and ERK signaling
By: Choi, Joo-Hee; Jang, Ah-Ra; Jeong, Ha-Na; Kim, Kiok; Kim, Young-Min; Cho, Jeong-Yong; Park, Jong-Hwan
International Journal of Medical Sciences (2020), 17(5), 632-639 | Language: English, Database: CAplus and MEDLINE

4-4. Tendril extract of Cucurbita moschata suppresses NLRP3 inflammasome activation in murine macrophages and human trophoblast cells
By: Park, Ji-Yeon; Jo, Sung-Gang; Lee, Ha-Nul; Choi, Joo-Hee; Lee, Yeon-Ji; Kim, Young-Min; Cho, Jeong-Yong; Lee, Sung Ki; Park, Jong-Hwan
International Journal of Medical Sciences (2020), 17(8), 1006-1014 | Language: English, Database: CAplus and MEDLINE

4-5. Dehydrodiconiferyl Alcohol Inhibits Osteoclast Differentiation and Ovariectomy-Induced Bone Loss through Acting as an Estrogen Receptor Agonist
By: Lee, Wonwoo; Ko, Kyeong Ryang; Kim, Hyun-keun; Lee, Doo Suk; Nam, In-Jeong; Lim, Seonung; Kim, Sunyoung
Journal of Natural Products (2018), 81(6), 1343-1356 | Language: English, Database: CAplus and MEDLINE

4-6. Effective suppression of pro-inflammatory molecules by DHCA via IKK-NF-κB pathway, in vitro and in vivo
By: Lee, Junghun; Choi, Jinyong; Kim, Sunyoung
British Journal of Pharmacology (2015), 172(13), 3353-3369 | Language: English, Database: CAplus and MEDLINE

Author Response

  1. The author identified the active substance of the leaf of Cucurbita moschataextract by looking at the function related to the promotion of bone formation.
  2. Readers want to know what substances are in fractions 6 and 7. To refer to the activity of fractions 6 and 7, please identify the compounds in fractions 6 and 7.

Authors Response: For the Reviewer only, we include the following data:

- Fractions 6 and 7 showed only one major spot by TLC analysis using dichloromethane:methanol (9:1) as the eluent.

- Crystallization from water provided 2 pure compounds that were subsequently identified as ferulic acid from fraction 6 and p-cumaric acid from fraction 7. 

- Data obtained by 1H and 13C NMR analysis for compounds purified from fractions 6 and 7 fully match those obtained from commercially available pure samples.

Considering that the highest effect was observed for the fraction 2, we prefer not include in the text the data about fractions 6 and 7.

  1. Add the NMR and MS data of compound 2 to the Supplementary Materials.

Authors Response: The required data are reported in the new version of Supplementary Figure S2.

  1. Recently, research on Cucurbita moschatathat is good for bones is continuously in progress (some mention in the introduction)

Authors Response: We agree with the Reviewer and we have mentioned what Reviewer requested in the Introduction adding some of the references he/she suggested.

Reviewer 2 Report

Thank you for the opportunity to review the submitted manuscript.

While the research seems to be done well with relevant models, the use of extracted material with acetone etc is hardly physiological and I doubt the relevance of this research to humans with osteoporosis. Most of the nutritional investigations recently look at whole foods, not fractions, as it is impossible to create a food including some fractionated ingredient.

Abstract: The abstract does not cover the results sufficiently. Lines 19 and 20 do not reflect the research but are non-scientific comments. What is meant by deepening of the therapeutic potential? Or enhancing a little studied part of the plant?

In the introduction the authors mention that they were exploring the pharmacological potential of the leaves; it is doubtful that a singe component extracted from a leaf would become a drug; most research currently focus on whole foods not bioactives.

Methods:

0.5% for DMSO is rather high and can exert effects by itself on cell proliferation.

DMSO as low as 0.35% enhances ALP activity and mineralisation in OBs: https://pubmed.ncbi.nlm.nih.gov/21652706/

DMSO as low as 0.2% induces proliferation, and as low as 0.5% induces osteoblastic differentiation, in MC3T3 cells: https://www.sciencedirect.com/science/article/pii/S001457930501433X

The comparator is 0.5% DMSO but what would the final concentration of DMSO be in each well?

Why was TLC used and not HPLC which would be more accurate to fractionate the samples?

Was the concentration of the fractions used in the cell culture by weight?

What is the relevance of the scratch-wound healing assay to osteoblasts?

What was the positive control for the ALP assay?

Do osteoblasts form actin rings?

With regards to the osteoclast assays, the usual method is to measure TRAP activity and then count the large multinucleated cells.

In figure 6c, the authors mention that the fatty acid decreased actin ring structures, but the graph indicates a higher percentage of actin ring positive cells?

Results and discussion:

I suggest that the research is repeated using the extracted fractions for osteoclastogenesis and then fractions etc. The fatty acid results in figure 6 are hardly of relevance to humans?  A fatty acid cannot be used in a real setting due to toxicity and oxidation.

The most effective treatments for osteoporosis currently are anti-resorptives; therefore, exploring the effects of the pumpkin leaves on osteoclastogenesis and function may be more of relevance.

In line 479 the authors mention nutritional agents; the extracts are not nutritional but synthetic.

Lines 481-484: The role of fatty acid in preventing bone loss is not emerging, there is a large body of work published for at least 20 years on the role of long chain fatty acids in bone health.

The authors used primary cell cultures to investigate the effects of the fractions: it may be easier to use cell lines such as the murine pre-osteoblast MC3T3-E1 or the murine macrophage RAW 264.7 for screening and then do further work using primary cells.

Author Response

Thank you for the opportunity to review the submitted manuscript.

While the research seems to be done well with relevant models, the use of extracted material with acetone etc is hardly physiological and I doubt the relevance of this research to humans with osteoporosis. Most of the nutritional investigations recently look at whole foods, not fractions, as it is impossible to create a food including some fractionated ingredient.

Authors Response: As reported in Discussion, our work mainly proposes a potential benefit from pumpkin leaves consumption or the possibility to extract from pumpkin leaves molecules with beneficial effects for bone health. Our experimental approach based on extracts and fractions is, in our opinion, the essential step for setting up informative in vitro experiments. On the other hand, this is not claimed as a nutritional study, but rather a preliminary screening of pumpkin leaf extract therapeutic potential by a bioassay-guided fractionation.

As regards the fact that the synergistic action of the various components in the raw extracts produces a more significant overall effect than the single fractions, we agree.

Abstract: The abstract does not cover the results sufficiently. Lines 19 and 20 do not reflect the research but are non-scientific comments. What is meant by deepening of the therapeutic potential? Or enhancing a little studied part of the plant?

Authors Response: We thank the Reviewer for his/her comments which give us the opportunity to improve the Abstract. In particular, the statement “The main objectives were aimed both at deepening the therapeutic potential of the plant in the field of skeletal health that has not yet been explored today, and enhancing a little studied part of the plant, the leaves, which in the West is considered a waste product.” has been replaced with “The data here obtained broaden the knowledge on the biological activities of the Cucurbita moschata and in particular underline the potential benefit of consuming leaves which are a part of the plant currently little considered in the Western world.”.

The sentence “…This dual effect was monitored by analyzing Runx2 expression, deposition of mineralized matrix, ALP activity, TRAP and actin ring staining respectively. “  has been added.

In the introduction the authors mention that they were exploring the pharmacological potential of the leaves; it is doubtful that a singe component extracted from a leaf would become a drug; most research currently focus on whole foods not bioactives.

Authors Response: We are sorry, but we do not agree with the Reviewer who may not have correctly interpreted the sentence given in the Introduction which refers to the work of other research groups; “In order to better define the pharmacological potential, some research groups are working on phytochemical screening of the content of leaf extracts from different cultivars.” ….

Methods:

0.5% for DMSO is rather high and can exert effects by itself on cell proliferation.

DMSO as low as 0.35% enhances ALP activity and mineralisation in OBs: https://pubmed.ncbi.nlm.nih.gov/21652706/

DMSO as low as 0.2% induces proliferation, and as low as 0.5% induces osteoblastic differentiation, in MC3T3 cells: https://www.sciencedirect.com/science/article/pii/S001457930501433X

Authors Response: We understand the concern of the reviewer. However, we would like to explain that:

  1. the percentage of DMSO used in the experiments is absolutely in line with many other works.
  2. with respect to the point raised relatively to proliferation and enhancement of mineral matrix deposition, in all the reported experiments, sample treated with DMSO was added as a control.
  3. DMSO 0.5 % is the maximum percentage we used, however the most of the experiments were performed with DMSO ≤ 0.1%. In order to preserve the accuracy of the experimental approach we have still considered 0.5% as the appropriate concentration to which we can refer.

The comparator is 0.5% DMSO but what would the final concentration of DMSO be in each well?

Authors Response: 0.5% (vol/vol), so 0.5 μl of neat DMSO (CAS –No. 67-68-5) in 100 μl of culture medium.

Why was TLC used and not HPLC which would be more accurate to fractionate the samples?

Authors Response: We used TLC because we are not currently able to handle a semipreparative and /or preparative HPLC (Genovese S, Taddeo VA, Epifano F, Fiorito S, Bize C, Rives A, de Medina P. Characterization of the Degradation Profile of Umbelliprenin, a Bioactive Prenylated Coumarin of a Ferulago Species J Nat Prod. 2017 Sep 22;80(9):2424-2431).

Was the concentration of the fractions used in the cell culture by weight?

Authors Response: All the organic extracts and fractions were weighted and were dissoved in neat dimethylsulfoxide (DMSO), to a final concentration of 50 mg/mL for Hexane and methanol extracts and 30 mg/mL for Acetone extracts, and diluted in culture medium to the working solution before use. This information was added in the Methods.

What is the relevance of the scratch-wound healing assay to osteoblasts?

Authors Response: It is well known that the osteoblasts migration is one of the key steps of bone regeneration/repair process. Since the ability to repair bone damage is an important feature in tissue regeneration, the quantification of this ability through scratch-wound healing assay is an important issue for the assessment of the pro-anabolic properties of biomolecules. Starting from this consideration we thought it was essential to include this experiment to evaluate the biological effect of the Cucurbita moschata extracts.

What was the positive control for the ALP assay?

Authors Response: In all experiments we measured ALP activity by using p-nitrophenyl phosphate disodium hexahydrate (pNPP) reagent, and not an Alkaline Phosphatase Assay Kit (Colorimetric). In order to satisfy the request of the Reviewer, new experiments would have to be performed which would take longer than the time allowed by the Editor for the re-submission of the manuscript. However, to meet the request of the Reviewer we have added in the experiments the value of negative controls represented by the same cells cultured in non osteogenic medium (basal medium).

Do osteoblasts form actin rings?

Authors Response: The formation of actin filament superstructure can be associated to several biological process, such as cytokinesis.  In particular actin ring is a characteristic actin structure that is essential for bone resorption by osteoclasts (we added this reference: Osteoclast Multinucleation: Review of Current Literature. Kodama J, Kaito T.Int J Mol Sci. 2020 Aug 8;21(16):5685). This particular cytoskeleton organization can be highlighted through Phalloidine conjugated staining. In order to clarify this special feature, we performed F-actin staining both in human osteoclasts (Figure 6C) and in osteoblasts (Figure 1C). As it can be observed, actin ring is specific for mature osteoclasts whilst osteoblasts didn’t display this organization. The extract – treated osteoblasts maintain a normal cytoskeleton as specified in the revised version.

With regards to the osteoclast assays, the usual method is to measure TRAP activity and then count the large multinucleated cells.

Authors Response: We thank the Reviewer for this observation which allow us to better explain the experiment. As reported in the legend of Figure 6 and in the Methods, the TRAP-positive cells were multinucleated cells containing more than three nuclei (this was added in the text and in the revised version of Figure 6 B). In any case, we would like to point out that we are using primary cells and not a cell line.

In figure 6c, the authors mention that the fatty acid decreased actin ring structures, but the graph indicates a higher percentage of actin ring positive cells?

Authors Response: The percentage of actin ring positive cells observed in human osteoclast after PU-13OH-FA treatments was significantly lowered respect to DMSO treated cells (P≤0.001). This reduction was also observed after treatment with acetone extract and its subfraction 2 (Fr2), with no significant difference between group ((P>0.05, ANOVA followed by Tukey’s post-hoc test was applied).

Results and discussion:

I suggest that the research is repeated using the extracted fractions for osteoclastogenesis and then fractions etc. The fatty acid results in figure 6 are hardly of relevance to humans?  A fatty acid cannot be used in a real setting due to toxicity and oxidation.

Authors Response: as it mentioned in the title of the Manuscript we focused on the anabolic effect of Violina pumpkin (Cucurbita moschata) Leaf Extracts. Therefore, the experiments were performed firstly on human primary osteoblasts. To confirm our hypothesis, the latest experiments were carried out on human osteoclasts, focusing only on what affected the osteoblasts (Acetone extract, TLC subfraction 2 (Fr2) and the single PU-13OH-FA compound). However, the suggestion of the Reviewer is correct. In fact, it is our intention to perform experiments with pumpkin leaf extracts not only on osteoclasts from healthy volunteers (as carried out here) but also on osteoporotic patients (work in progress). For what concerns the use of a fatty acid in our study, as stated in the response on previous comment, our approach is a bioassay guided fractionation investigation on a vegetable extract and not a nutritional study.

In any case, the health effects of dietary unsaturated fatty acids are widely studied. We are aware of the caution and special attention in monitoring the behaviour of lipid-treated cells, however, in our opinion, the data we have obtained may be relevant for humans. The results here reported can be considered preliminary to a future broader and more detailed analysis, taking into account numerous studies in the literature on viability, gene expression and differentiation performed in different cell types after treatment with fatty acids.

Some references:

- Potential modulatory mechanisms of action by long-chain polyunsaturated fatty acids on bone cell and chondrocyte metabolism. Abshirini M, Ilesanmi-Oyelere BL, Kruger MC. Prog Lipid Res. 2021 Jul 2;83:101113

- Cell culture models of fatty acid overload: Problems and solutions. Alsabeeh N, Chausse B, Kakimoto PA, Kowaltowski AJ, Shirihai O. Biochim Biophys Acta Mol Cell Biol Lipids. 2018 Feb;1863(2):143-151.

- The effect of trans-palmitoleic acid on cell viability and sirtuin 1 gene expression in hepatocytes and the activity of peroxisome-proliferator-activated receptor-alpha. Farokh Nezhad R, Nourbakhsh M, Razzaghy-Azar M, Sharifi R, Yaghmaei P. J Res Med Sci. 2020 Nov 26;25:105

-  Schulze-Tanzil G ,  de SP,  Behnke B,  Klingelhoefer S,  Scheid A,  Shakibaei M. Effects of the antirheumatic remedy hox alpha--a new stinging nettle leaf extract--on matrix metalloproteinases in human chondrocytes in vitro. Histol Histopathol.2002 Apr;17(2):477-85.

- Kasonga AE ,  Deepak V ,  Kruger MC ,  Coetzee M . Arachidonic acid and docosahexaenoic acid suppress osteoclast formation and activity in human CD14+ monocytes, in vitro. PLoS One. 2015 Apr 13;10(4):e0125145.

- Rahman MM,  Bhattacharya A,  Fernandes G. Docosahexaenoic acid is more potent inhibitor of osteoclast differentiation in RAW 264.7 cells than eicosapentaenoic acid. J Cell Physiol. 2008 Jan;214(1):201-9.

- Kumar N,  Gupta G,  Anilkumar K,  Fatima N,  Karnati R,  Venkateswara Reddy G ,  Giri PV, Reddanna P. 15-Lipoxygenase metabolites of α-linolenic acid, [13-(S)-HPOTrE and 13-(S)-HOTrE], mediate anti-inflammatory effects by inactivating NLRP3 inflammasome. Sci Rep.2016 Aug 18;6:31649

The most effective treatments for osteoporosis currently are anti-resorptives; therefore, exploring the effects of the pumpkin leaves on osteoclastogenesis and function may be more of relevance.

Authors Response: We are sorry, but we do not completely agree with the Reviewer. The antiresorptive therapy is as important as anabolic therapy; in many cases it is assumed that the stimulation of bone formation is more favourable to improve bone regeneration than the inhibition of bone resorption.

In line 479 the authors mention nutritional agents; the extracts are not nutritional but synthetic.

Authors Response: We delete “nutritional agents” and added the following statement: “Although more studies are needed, however, our data suggests that consuming leaves may be nutritionally beneficial for bone health.”

Lines 481-484: The role of fatty acid in preventing bone loss is not emerging, there is a large body of work published for at least 20 years on the role of long chain fatty acids in bone health.

Authors Response: We agree with the Reviewer and we are sorry for the mistake. “is emerging” has been replaced with “have long been studied”.

The authors used primary cell cultures to investigate the effects of the fractions: it may be easier to use cell lines such as the murine pre-osteoblast MC3T3-E1 or the murine macrophage RAW 264.7 for screening and then do further work using primary cells.

Authors Response: We agree with the Reviewer: working with cell lines is certainly easier than with primary cell cultures. However, as we mentioned in the Discussion “It is important to underline that, performing the experiments with human primary bone cell cultures certainly has an added value over the use of cell lines as they resemble the cell behavior in vivo.” In agreement with a large part of the scientific community, we are firmly convinced of the need to adopt adequate human experimental models which, trying to mimic the physiological microenvironment, are as informative as possible. Therefore, considering that the effects of any in vitro treatment may be dependent on cell types and are very often different in mouse cells than in human cells, we preferred to focus our attention directly on human primary bone cells.

Round 2

Reviewer 1 Report

According to the author's explanation, the compounds of 6 and 7 are well-known simple compounds, and they are compounds of a different structure than the second compound. Therefore, it is judged as better data because the second compound is emphasized. Please add the spectroscopic (NMR) data of compounds 6 and 7 to the supplementary information and add the relevant information to the manuscript.

S2 data does not contain raw data. Please enter raw data so that you can check the purity of the compound (Please show the data that the activity result of compound 2 is the activity of the compound, not the complex compound)
If it is difficult to confirm the purity through NMR or HPLC, it seems that TLC can sufficiently show it.

If the place represented by fr.2 & 3 in Figures 4, 5 and 6 was an experiment using a compound, please enter the name of the compound whose structure was determined.

Author Response

Reviewer 1.

According to the author's explanation, the compounds of 6 and 7 are well-known simple compounds, and they are compounds of a different structure than the second compound. Therefore, it is judged as better data because the second compound is emphasized. Please add the spectroscopic (NMR) data of compounds 6 and 7 to the supplementary information and add the relevant information to the manuscript.

Authors Response: NMR data of fractions 6 and 7 together with adequate comment and references have been added in the revised version of Supplemental Figure 2 C. The sentence “Characterization of compounds present in the fractions 6 and 7 are reported in the Supplemental Figure 2 C” has been added in the manuscript. We are convinced to provide the information regarding these fractions as supplementary information as their inclusion in the Results does not add relevant data to purpose of the Paper and would distract from the focus that we have chosen to do on fraction 2. However, we agree with the Reviewer on the importance of providing the data he/she required.

S2 data does not contain raw data. Please enter raw data so that you can check the purity of the compound (Please show the data that the activity result of compound 2 is the activity of the compound, not the complex compound)
If it is difficult to confirm the purity through NMR or HPLC, it seems that TLC can sufficiently show it.

Authors Response: As stated in paragraph 2.3 of the revised manuscript purity of PU-13OH-FA was assessed by GC-MS and derivatization of the acid as methyl ester. This is rather a common way to establish purity of compounds, surely more effective than TLC. Both in the text and in the Supplemental Figure 2, we have added the value of purity for the compound.

We think that Results and Figures are well described: Fractions were prepared as described in the Methods and PU-13OH-FA was purified from Fraction 2; their activity were compared.

If the place represented by fr.2 & 3 in Figures 4, 5 and 6 was an experiment using a compound, please enter the name of the compound whose structure was determined.

Authors Response: This section of the manuscript does not refer to a pure compound.

Reviewer 2 Report

The authors have addressed all of the queries, no further comments.

Author Response

Reviewer 2.

Authors Response: Thank you for your positive comments.